# *Vibrio cholerae* requires oxidative respiration through the *bd*-I and *cbb*$_3$ oxidases for intestinal proliferation

**Andrew J. Van Alst**[¤], **Lucas M. Demey, Victor J. DiRita**\*

Department of Microbiology and Molecular Genetics, Michigan State University, East Lansing, Michigan, United States of America

¤ Current address: Department of Molecular & Cell Biology, University of California, Berkeley, Berkeley, California, United States of America

\* diritavi@msu.edu

**Data Availability Statement:** Data are available at Zenodo repository DOI: 10.5281/zenodo.6342667, an Excel spreadsheet containing, in separate sheets, the underlying numerical data and

## Abstract

*Vibrio cholerae* respires both aerobically and anaerobically and, while oxygen may be available to it during infection, other terminal electron acceptors are proposed for population expansion during infection. Unlike gastrointestinal pathogens that stimulate significant inflammation leading to elevated levels of oxygen or alternative terminal electron acceptors, *V. cholerae* infections are not understood to induce a notable inflammatory response. To ascertain the respiration requirements of *V. cholerae* during infection, we used Multiplex Genome Editing by Natural Transformation (MuGENT) to create *V. cholerae* strains lacking aerobic or anaerobic respiration. *V. cholerae* strains lacking aerobic respiration were attenuated in infant mice 10$^5$-fold relative to wild type, while strains lacking anaerobic respiration had no colonization defect, contrary to earlier work suggesting a role for anaerobic respiration during infection. Using several approaches, including one we developed for this work termed Comparative Multiplex PCR Amplicon Sequencing (CoMPAS), we determined that the *bd*-I and *cbb*$_3$ oxidases are essential for small intestinal colonization of *V. cholerae* in the infant mouse. The *bd*-I oxidase was also determined as the primary oxidase during growth outside the host, making *V. cholerae* the only example of a Gram-negative bacterial pathogen in which a *bd*-type oxidase is the primary oxidase for energy acquisition inside and outside of a host.

## Author summary

The bacterium that causes cholera, *Vibrio cholerae*, can grow with or without oxygen. When growing without oxygen it may use other molecules that serve the same purpose as oxygen, acting as a terminal electron acceptor in an energy-generating process known as respiration. Given the largely anaerobic nature of the gastrointestinal tract, and the lack of significant inflammation during cholera infection, a process that can stimulate elevated levels of oxygen and other terminal electron acceptors, we sought to understand the respiratory mechanisms of *V. cholerae* during infection. We used a powerful genome-editing

statistical analysis for Figs panels 2B–2D, 3A–3E, 4A–4D, 5A–5D, 6A–6C, 7A–7B, S1A–S1C, S3A–S3D, S4A–S4B, S5A–S5D, S6A–S6G, S7B–S7D, and S8A–S8F.

**Funding:** This work was supported in part by the Rudolph Hugh Endowment (V.J.D.) at Michigan State University. A.J.V. was supported in part by the Eleanor L. Gilmore Endowed Excellence Award in the Department of Microbiology and Molecular Genetics at Michigan State University. The funders had no role in study design, data collection and analysis, decision to publish, or preparation of the manuscript.

**Competing interests:** The authors have declared that no competing interests exist.

method to construct mutant strains of *V. cholerae* lacking some or all of the complement of proteins required for aerobic or anaerobic respiration. By analyzing these mutants in the laboratory and in intestinal colonization of infant mice, we determined that the ability to respire without oxygen is completely dispensable for *V. cholerae* to thrive during infection. We determined that two of the four oxygen-dependent respiration mechanisms are essential for *V. cholerae* to grow during infection, with the other two dispensable for wild type levels of colonization.

## Introduction

Respiration promotes growth and proliferation of bacterial cells [1]. Energy acquisition through respiration relies on the metabolism of exogenously acquired substrates and the presence of terminal electron acceptor molecules to generate chemical energy, which is stored in the form of ATP to power cellular processes required for growth. Although not considered canonical virulence factors, metabolism and energy generative processes are required by pathogens to thrive during infection. Here we investigate respiration as a potent driver of replication during infection by the bacterial gastrointestinal pathogen *Vibrio cholerae* [2].

*Vibrio cholerae* is a facultative anaerobe that grows in both aerobic and anaerobic environments [3]. Respiration in *V. cholerae* is achieved aerobically through the terminal reduction of molecular oxygen or anaerobically through the terminal reduction of various alternative electron acceptors [4,5]. Recent evidence suggests there is combined contribution of both aerobic and anaerobic metabolism to *V. cholerae* growth *in vivo* [6,7]. This may be attributed to the radial and longitudinal gradients of oxygen availability in the intestinal tract enabling metabolism through both pathways in response to *in vivo* localization [8].

We sought to investigate the relative contributions of aerobic and anaerobic respiration *in vivo* using an infant mouse model of colonization. *V. cholerae* encodes four respiratory terminal oxidases and four terminal reductases that support respiration [9,10]. Terminal oxidases are enzyme complexes that transfer electrons to oxygen in the final step of the electron transport chain. Terminal reductases perform the same function of electron transfer, however, electrons are transferred to substrates other than oxygen. The terminal oxidases include one $cbb_3$ heme-copper oxidase [11,12] and three *bd*-type oxidase complexes capable of catalyzing the 4 $H^+/O_2$ reduction of oxygen to water [13]. $cbb_3$ oxidases [14,15] and *bd* oxidases [16–18] have a low $K_m$ for oxygen and are typically induced under microaerobiosis, a feature particularly beneficial for pathogens colonizing near hypoxic environments of the human host [19–21]. *V. cholerae* also carries out anaerobic respiration through four terminal reductases that use either nitrate, fumarate, trimethylamine-N-oxide (TMAO) or biotin sulfoxide (BSO) [5,10,22,23]. Previous work found that abrogation of nitrate reductase activity reduced colonization in a streptomycin-treated adult mouse by approximately 2-fold [4] and a concomitant disruption in fermentative pathways further reduced colonization, revealing a dependency between nitrate reduction and fermentation [7]. Additionally, TMAO influences virulence gene expression in *V. cholerae* when added exogenously, however, whether reduction of TMAO is required for this response is not clear [5,24]. In this study, we looked to more thoroughly examine the complete suite of terminal electron accepting complexes encoded by *V. cholerae* and assess the pertinence of each terminal electron acceptor molecule to *in vivo* infection.

By targeting terminal oxidase and terminal reductase complexes of *V. cholerae*, we can better understand the respiratory processes occurring during disease and identify which terminal electron accepting complexes are most critical to *in vivo* fitness. This work highlights that

oxygen, although present at low levels diffusing from the host epithelium, is sufficient and essential to supporting *V. cholerae* growth in the infant mouse. These findings change how we understand the host environment during *V. cholerae* pathogenesis and how oxygen may function in the pathogenicity of other bacterial pathogen elicited diseases.

## Results

### Constructing terminal electron acceptor mutant strains

*Vibrio cholerae* terminal electron acceptor mutants were generated using the multiplex genome editing technique MuGENT [25] and by positive allelic exchange vector pKAS32 [26] in an El Tor C6706 *V. cholerae* background. Target loci and their relative chromosomal locations are depicted in Fig 1A. In MuGENT-generated mutant strains, target loci are disrupted by a frameshift mutation, removal of ATG start codon, insertion of 3-frame stop codons, and offsetting of the ribosomal binding site. This is combined with the insertion of a universal detection sequence at each target locus and a spectinomycin cassette insert into pseudogene VC1807 for naturally competent cell selection that has no *in vitro* fitness cost [27] (S1 Fig.). MuGENT generated strains are designated with a superscript 'Mu' ($^{Mu}$). Mutant strains constructed via pKAS32 have the complete coding sequence for all subunits of target terminal electron acceptor complexes excised, generating isogenic deletion strains. Select strains were verified by whole genome sequencing which indicated no nucleotide polymorphisms in most strains and where present were found in hypothetical protein regions of the genome predicted to have no impact on bacterial cell fitness (S1 Table).

 *V. cholerae* encodes one $cbb_3$ oxidase and three *bd*-type oxidase complexes [9]. Cytochrome oxidase $cbb_3$ is a four subunit (VC1439-VC1442) cytochrome *c* containing terminal oxidase of which the coding sequence for the primary subunit, CcoN (VC1442), was disrupted to generate MuGENT knockout strains. CcoN is the first open reading frame in the operon and contains the active site for reducing oxygen to water [28]; its disruption was sufficient for abolishing $cbb_3$ cytochrome *c* activity (S2A Fig). For each of the three *bd*-type oxidases, *bd*-I (VC1844-43), *bd*-II (VCA0872-73), and *bd*-III (VC1570-71), both subunits of each complex were disrupted. Mutations in each target locus were confirmed by multiplex allele-specific PCR (MASC-PCR) [29] where the presence of a DNA band indicates successful genomic editing at the indicated locus via MuGENT (Fig 1B). To further validate the function of each oxidase, isogenic deletion strains were also generated for select terminal oxidases via pKAS32 positive allelic exchange.

 *V. cholerae* also encodes four alternative terminal reductase complexes [9] capable of reducing alternative electron acceptors that can support respiration in the absence of oxygen. The four terminal reductases include a fumarate reductase (VC2656-59), trimethylamine-N-oxide (TMAO) reductase (VC1692-94), nitrate reductase (VCA0676-0680), and a biotin sulfoxide reductase (BSO) (VC1950-51). For each of these multi-subunit complexes, the active reducing subunit was disrupted via MuGENT and confirmed by MASC-PCR (Fig 1B).

### *In vitro* characterization of terminal electron acceptor complex mutants

**Terminal oxidase growth characterization.** *V. cholerae* oxidase mutant strains were grown in LB media in aerobic and anaerobic conditions. Inocula were prepared anaerobically to ensure consistent growth of oxidase-deficient strains. After anaerobic preparation of inocula, both $cbb_3$ and *bd*-I oxidase complexes were found to be required for wild type levels of growth in aerobic conditions whereas *bd*-II and *bd*-III oxidases were not (Fig 2A). Cultures lacking the $cbb_3$ oxidase grew at a consistently lower optical density and never reached the peak $OD_{600}$ of wild type. The *bd*-I oxidase was determined to be the most critical oxidase

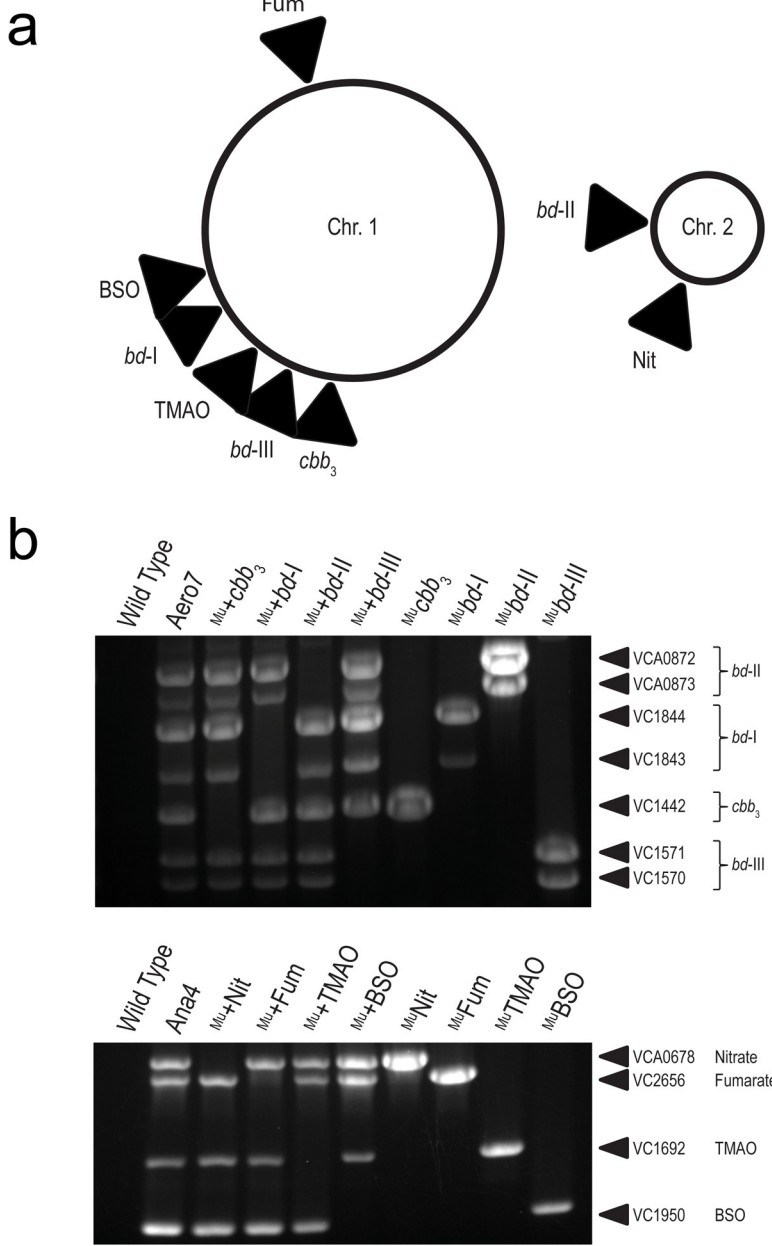

**Fig 1. Verification of MuGENT generated mutant strains.** (a) Chromosomal map of *V. cholerae* terminal electron reducing complex loci. (b) Multiplex allele-specific PCR (MASC-PCR) of *V. cholerae* terminal electron reducing complexes MuGENT mutants. Lanes are labelled with the strain name where a strain preceded by a 'Mu+' (Lanes 3–6) indicates the oxidase complex as the sole remaining functional oxidase in that strain and strains preceded by a 'Mu' (Lanes 7–10) indicates that the specified locus is the targeted knock out. Targeted gene loci are labelled to the right of each gel image. The presence of a band indicates a targeted knockout in the gene locus whereas the absence of a band indicates the wild type gene is present.

complex for supporting aerobic respiration in *V. cholerae*, as cells lacking it showed drastically reduced growth. This finding was unexpected as electron transport and oxygen reduction by the $cbb_3$ oxidase is more efficient at generating a proton gradient (and therefore ATP) for the cell [30]. Electrons passed to the $cbb_3$ oxidase are shuttled through the cytochrome $bc_1$ complex, which accounts for a $\Delta 6H^+$ proton gradient [31] along with translocation of two

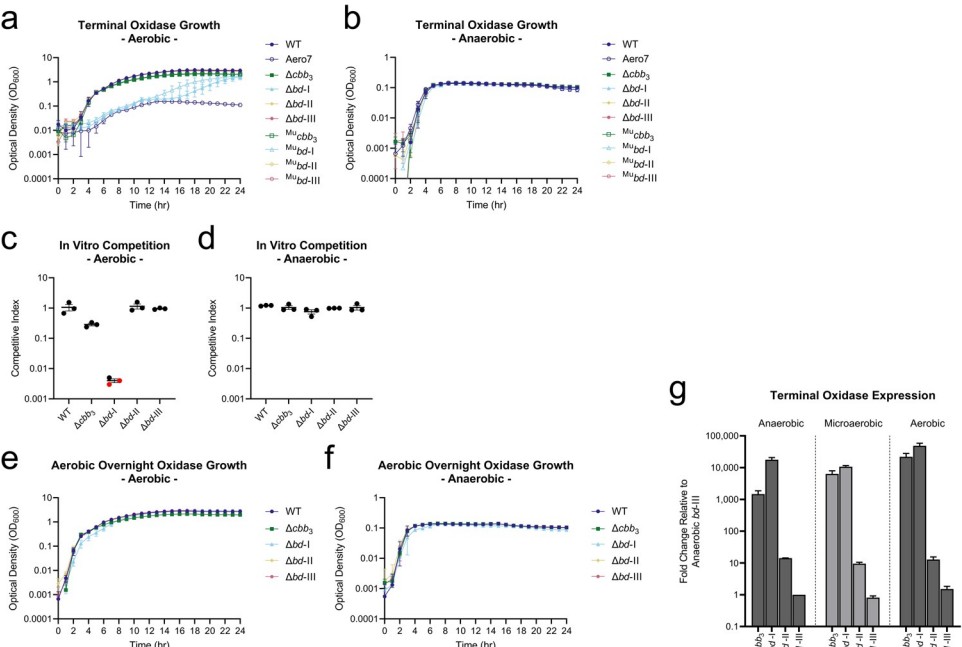

**Fig 2. Terminal oxidases support aerobic growth in *V. cholerae*.** Growth characteristics of the terminal oxidases in *V. cholerae*. (a-b) Single terminal oxidase mutants, both MuGENT and isogenic deletion, growth in LB. Inoculums were prepared anaerobically and subsequently grown in aerobic and anaerobic conditions, respectively. (c-d) Single terminal oxidase isogenic deletion strain *in vitro* LB competition assays in both aerobic and anaerobic conditions, respectively. Competitive index scores were calculated as a ratio of output versus input [(Target$_{Output}$/$\Delta lacZ_{Output}$) / (Target$_{Input}$/$\Delta lacZ_{Input}$)], where a $\Delta lacZ$ strain served as a psuedo-wild type to determine relative fitness via blue-white screening. Red dots indicate the limit of detection where no target strain CFUs were recovered for these trials. (e-f) Single terminal oxidase isogenic deletion mutant growth in LB where inoculums were prepared aerobically and subsequently grown in aerobic and anaerobic conditions, respectively. (g) *In vitro* expression of terminal oxidases in anaerobic, microaerobic, and aerobic growth conditions.

additional protons coupled to the terminal reduction of oxygen by the $cbb_3$ complex [32]. The *bd*-I reducing pathway, as with all *bd*-type oxidases, generates a relatively weaker proton gradient resulting in less ATP for the cell [13,33]. Thus our observation that it serves as the primary oxidase in *V. cholerae* under atmospheric oxygen conditions was unanticipated. All oxidase-disrupted mutants grew comparably to wild type in anaerobic conditions via anaerobic fermentation (Fig 2B) suggesting the observed defect in aerobic growth is not due to a general growth defect imposed by the mutations. These growth phenotypes were recapitulated in M9 0.2% D-glucose media, although *bd*-I deficient strains were further hampered for growth aerobically and showed a minor shift in reaching exponential phase anaerobically (S3A and S3B Fig). *In vitro* competition assays were also performed, demonstrating a competitive defect for both $cbb_3$ and *bd*-I deficient strains in aerobic, but not anaerobic, conditions (Fig 2C and 2D). When grown separately, individual deletion mutants were recovered at comparable colony forming units (CFU), similar to what is observed at later timepoints in the growth curve assay. We therefore attribute the competitive defect of $\Delta bd$-I to the extended lag phase observed in the early stages of growth where wild type can rapidly take over the population and exclude $\Delta bd$-I growth (S4A and S4B Fig).

We hypothesized that *bd*-I deficient strains grown anaerobically may experience a growth lag prior to expression of alternative oxidases, such as the $cbb_3$ oxidase (S2B Fig), accounting for the observed growth kinetics. To test this, we prepared inocula aerobically and observed the growth phenotype. In this condition, strains lacking *bd*-I oxidase grew to wild type optical

density (Fig 2E) indicating that delayed expression of alternative oxidases when inocula are prepared anaerobically may account for the observed growth lag. Similar growth patterns were observed among all strains in anaerobic growth conditions for inocula of aerobically prepared oxidase mutants (Fig 2F). We examined expression patterns of wild type *V. cholerae* under anaerobic, microaerobic, and aerobic conditions and determined relative quantification of gene expression by an elongated Pfaffl method detailed in Materials and Methods. *recA* served as the gene of reference and expression values are reported relative to *bd*-III expression in anaerobic conditions which served as the comparator target for relative expression among the oxidases (Fig 2G). The *cbb*$_3$ and *bd*-I oxidases were highly expressed relative to *bd*-II and *bd*-III. In anaerobic conditions, the *bd*-I oxidase was expressed nearly 10-times higher than the *cbb*$_3$ oxidase, whereas in the presence of oxygen (microaerobic or aerobic), *bd*-I oxidase was expressed only marginally higher than *cbb*$_3$ oxidase. All strains with a functional *bd*-I oxidase grew comparable to wild type during the shift from anaerobic to aerobic conditions whereas Δ*bd*-I exhibited an extended lag phase that was ameliorated when inocula were prepared aerobically. The *bd*-I oxidase was the most abundantly expressed oxidase in anaerobic conditions, suggesting that accumulation of this oxidase in anaerobiosis readies *V. cholerae* for rapid growth during transition to aerobic atmospheric conditions. We hypothesize that expression and/or activity of the *cbb*$_3$ oxidase is delayed during this transition period as *bd*-I deficient *V. cholerae* gradually exit lag phase, increasing *cbb*$_3$ transcript abundance and exhibiting *cbb*$_3$ activity in oxidative environments (S2B Fig).

Further characterization of the oxidase complexes was carried out using double, triple, and quadruple oxidase mutant strains. Isogenic double and triple deletion strains, as well as quadruple oxidase MuGENT mutant Aero7, were grown in aerobic and anaerobic conditions in LB from inocula grown anaerobically (Fig 3A and 3B). Strain +*bd*-I, harboring solely a functional *bd*-I oxidase, grew to near wild type levels while strains +*cbb*$_3$ and +*bd*-III, having solely the *cbb*$_3$ or *bd*-III oxidase complex, grew after a considerable lag phase. This lag phase, however, was reduced when inocula were adapted to an aerobic environment prior to the growth assay (Fig 3C and 3D). Growth phenotypes were comparable to triple oxidase MuGENT strains, although strain $^{Mu}$+*bd*-III (encoding only *bd*-III) exited lag phase more rapidly (S3C and S3D Fig). Strain Aero7, defective for production of all terminal oxidases encoded by *V. cholerae*, and strain +*bd*-II, containing only the *bd*-II oxidase, were completely deficient for aerobic growth. This was further exemplified for Aero7 by a near $10^7$-fold attenuation in *in vitro* competition assays (Fig 3E). From these results, we conclude that *cbb*$_3$, *bd*-I, and *bd*-III oxidases support aerobic growth of *V. cholerae* to varying degrees while *bd*-II does not. Despite its low mRNA expression level in wild type cells relative to other oxidases, *bd*-III oxidase supported +*bd*-III aerobic growth, particularly when culture inocula were grown aerobically.

Taken together, these findings make clear that *bd*-I oxidase is the primary oxidase in *V. cholerae*, priming a transition from anaerobic to aerobic environments and functioning as the primary oxidase in atmospheric oxygen environments. To our knowledge, *V. cholerae* and *Listeria monocytogenes* [34] are the only pathogens demonstrated to preferentially use a *bd*-type oxidase in lieu of a heme-copper oxidase such as the *bo*$_3$ oxidase of *Escherichia coli* [35] and *Salmonella* Typhimurium [36], the *cbb*$_3$ oxidase of *Psuedomonas aeruginosa* [37], *Campylobacter jejuni, and Helicobacter pylori* [38], or the *aa*$_3$ oxidase of *Staphylococcus aureus* [39] to support growth in atmospheric oxygen.

**Growth of mutants lacking terminal reductases.** MuGENT terminal reductase *V. cholerae* mutants were prepared aerobically and used to inoculate fresh LB with and without alternative electron acceptors in both aerobic and anaerobic conditions. Individual reductase MuGENT-derived mutant strains were made lacking the active subunit of fumarate reductase (VC2656), TMAO reductase (VC1692), nitrate reductase (VCA0678), and BSO reductase

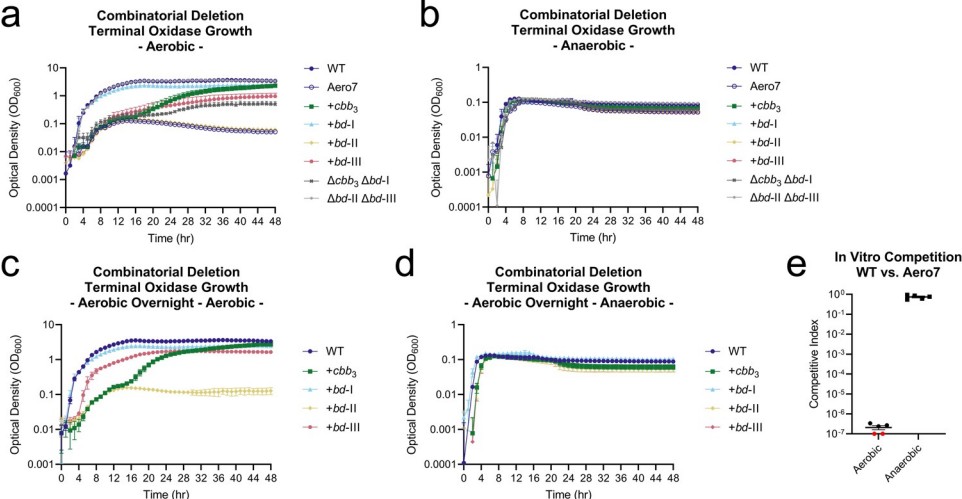

**Fig 3. Individually, oxidases $cbb_3$, $bd$-I, and $bd$-III support aerobic growth in *V. cholerae*.** (a-b) Combinatorial terminal oxidase deletion mutant growth in LB. Inoculums were prepared anaerobically and subsequently grown in aerobic and anaerobic conditions, respectively. Triple deletion mutant strains have a '+' with an oxidase name (e.g. $+cbb_3$), indicating the sole remaining oxidase, with the other three oxidases disrupted by mutation. (c-d) Combinatorial terminal oxidase deletion mutant growth in LB. Inoculums were prepared aerobically and subsequently grown in aerobic and anaerobic conditions, respectively. (e) *In vitro* aerobic and anaerobic competition assay between Aero7 and wild type *V. cholerae* with competitive index scores calculated as [(Aero7$_{Output}$/WT$_{Output}$) / (Aero7$_{Input}$/WT$_{Input}$)]. Growth curves are an average of three biological replicates where error bars represent the standard error of the mean. Bars for *in vitro* competitions and expression data represent the arithmetic mean where error bars represent the standard error of the mean.

(VC1950). A combinatorial mutant was also constructed, denoted Ana4, in which all reductases were disrupted.

All terminal reductase mutants, including Ana4, grew similarly to wild type in LB in both aerobic and anaerobic growth conditions. However, when grown anaerobically in the presence of the alternative electron acceptors fumarate, TMAO, nitrate, or DMSO, mutant strains were defective for growth compared to wild type (Fig 4A–4D). This indicated that the MuGENT-generated reductase mutants were defective for reductase function. As Ana4 was reduced for growth in the presence of all alternative electron acceptors, we concluded it adequately represented a strain incapable of utlizing these molecules to support growth via anaerobic respiration and included it in our *in vivo* analysis described below. The presence of these alternative electron acceptors under aerobic conditions led to varying growth responses by wild type *V. cholerae*. With fumarate or DMSO, growth was boosted, however, with either nitrate or TMAO growth was reduced (S5A–S5D Fig).

## Aero7 and Ana4 infant mouse infections

To examine the importance of aerobic and anaerobic respiration during infection, *V. cholerae* Aero7 and Ana4 MuGENT strains were tested for their ability to colonize the infant mouse intestinal tract. Single strain and competition infections in neonatal mice were performed for both strains.

Aero7 was severely attenuated for colonization of the small intestine, in both single strain and competition infections (Fig 5A and 5B). In single strain infections, wild type *V. cholerae* was recovered near $10^8$ CFU/g intestine whereas Aero7 was recovered near $10^3$ CFU/g intestine, a 5-log decrease in colonization. This reduction was also observed in the competition infections where the competitive index (CI) score of Aero7, calculated as [[Aero7$_{Output}$/

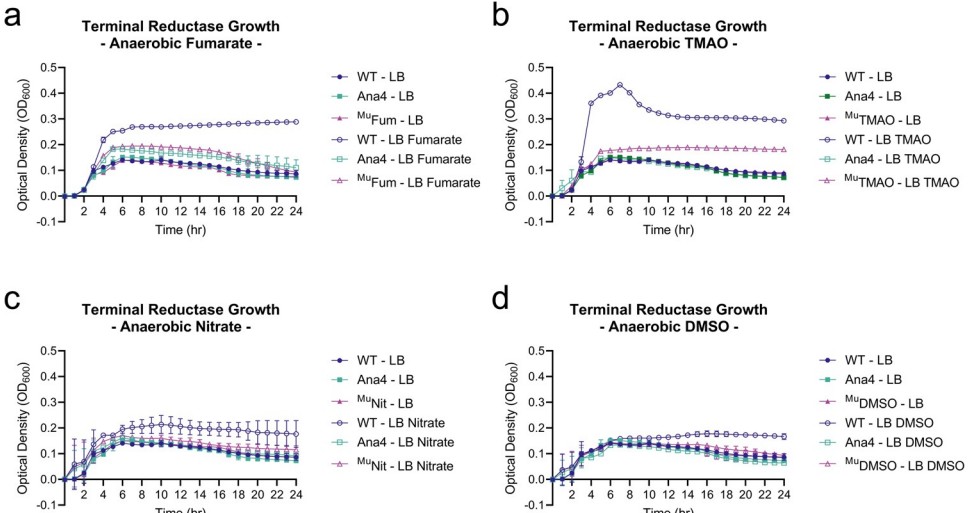

**Fig 4. Terminal reductase mutants are reduced for anaerobic growth in the presence of cognate electron acceptor molecules.** Growth characteristics of terminal reductases of *V. cholerae*. MuGENT generated terminal reductase mutants grown in LB in the presence and absence of alternative electron acceptors (a) 50mM fumarate, (b) 50mM trimethylamine-N-oxide (TMAO), (c) 50mM nitrate, and (d) 50mM dimethyl sulfoxide (DMSO). Inoculums were prepared aerobically and subsequently grown in anaerobic conditions. Growth curves are an average of three biological replicates where error bars represent the standard error of the mean.

$WT_{Output}$] / [$Aero7_{Input}$/$WT_{Input}$]] was approximately $10^{-5}$. Aero7 competed better in the large intestine, although was still at a fitness disadvantage, with a CI of $10^{-3}$. Its greater fitness in the large intestine may be due to the more anaerobic environment of this site (S6A and S6B Fig). That Aero7 is considerably less fit in the infant mouse small intestine is relevant as this is the site of infection in humans. We conclude from these findings that maintaining functional terminal oxidases, and therefore aerobic respiration, is critical for *V. cholerae* to establish infection and proliferate.

In contrast to a strain lacking all terminal oxidases, the Ana4 mutant lacking all terminal reductases, which is deficient for growth via anaerobic respiration (Fig 4A–4D), colonized both the small and large intestines to wild type levels in both single strain and competition infections (Figs 5C and 5D and S6C and S6D). The lack of an observable phenotype in our experiments differs from an earlier study demonstrating a two-fold reduction in colonization by a mutant strain of *V. cholerae* lacking nitrate reductase (*napA*; VCA0678) in the adult streptomycin-treated mouse model [4]. This two-fold reduction observed in adult mice, but not infant, may be related to the myriad conditions that impact the overall oxygenation levels (or lack thereof) in the mammalian gut [40], and which may differ between infant and adult mice. Additionally, nitrate availability has also been shown to increase following streptomycin treatment, which may contribute to the observed colonization discrepancy between wild type and *napA V. cholerae* [41], as lacking a functional nitrate reductase may incur a greater fitness cost in the streptomycin-treated adult mouse.

Overall, our data suggests that anaerobic respiration using alternative electron acceptors is not a prominent feature of *V. cholerae* growth during infection of the infant mouse. The slight growth defect reported with a nitrate reductase mutant in the streptomycin-treated adult mouse [4] suggests that there may be a limited role for anaerobic respiration although more work is required to ascertain the effects of a mature, non-disturbed microbiota on these questions.

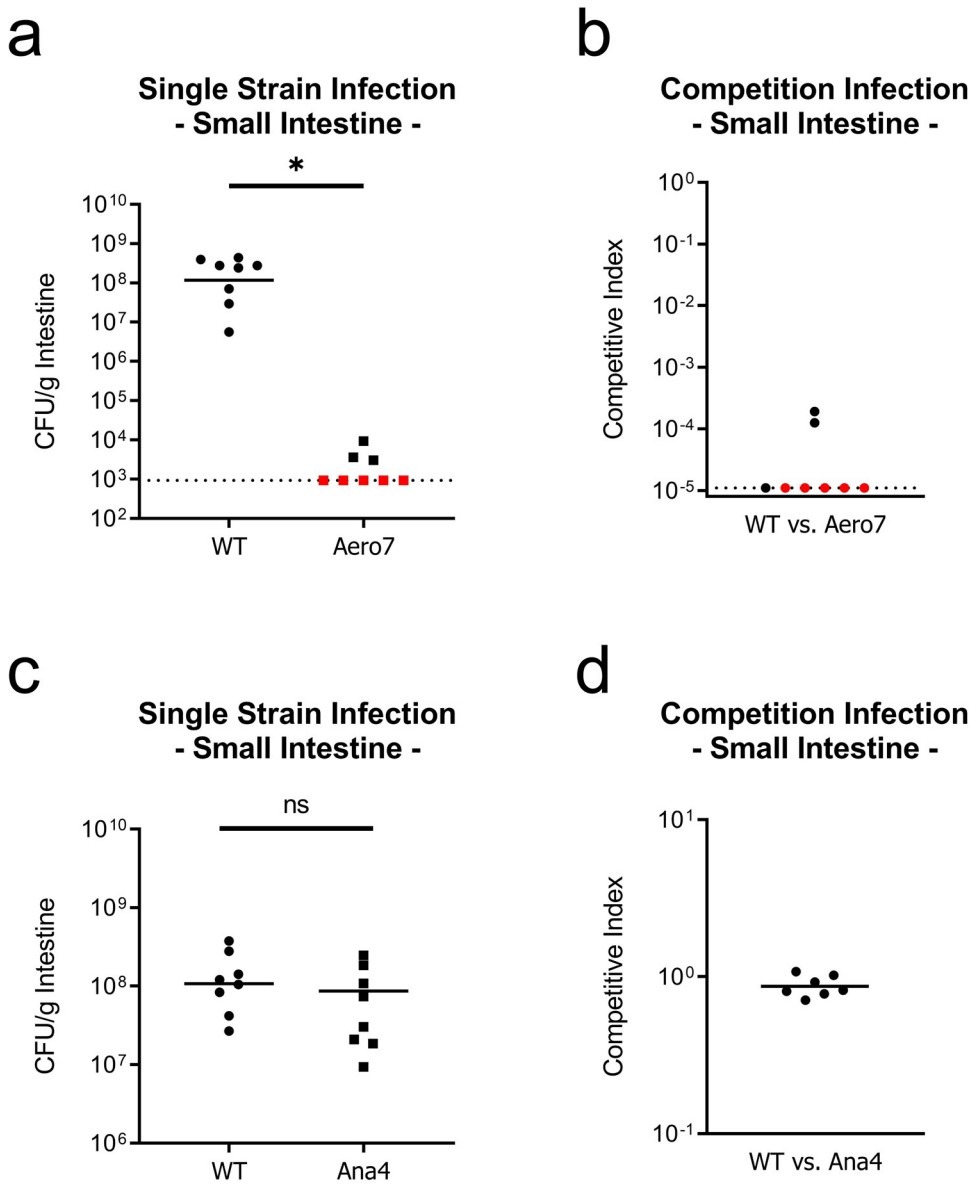

**Fig 5. Aerobic respiration, but not anaerobic respiration, is required for growth and colonization of the infant mouse small intestine.** Aero7 and Ana4 small intestine colonization in single strain and competition infections. (a) Single strain infection of strain Aero7. (b) Competition infection of strain Aero7. (c) Single strain infection of strain Ana4. (d) Competition infection of strain Ana4. Bars represent the geometric mean. Horizontal dashed lines indicate the limit of detection (LOD) and red dots indicate recovered CFUs were below the LOD. Competitive index scores were calculated as [(Mutant$_{Output}$/WT$_{Output}$) / (Mutant$_{Input}$/WT$_{Input}$)]. Statistical analysis was performed using GraphPad PRISM. *, $P < 0.05$. A Mann-Whitney U-test was used in the determination of significance between WT and Aero7. A Student's $t$ test was performed on log transformed data in the determination of significance between WT and Ana4.

## Individual oxidase function during infection

**Comparative Multiplex PCR Amplicon Sequencing (CoMPAS).**   To examine the requirements of the terminal oxidases of *V. cholerae in vivo*, we took a novel approach that combines elements of insertion-site sequencing (Tn-Seq) with targeted amplification of MuGENT generated oxidase mutations (Comparative Multiplex PCR Amplicon Sequencing

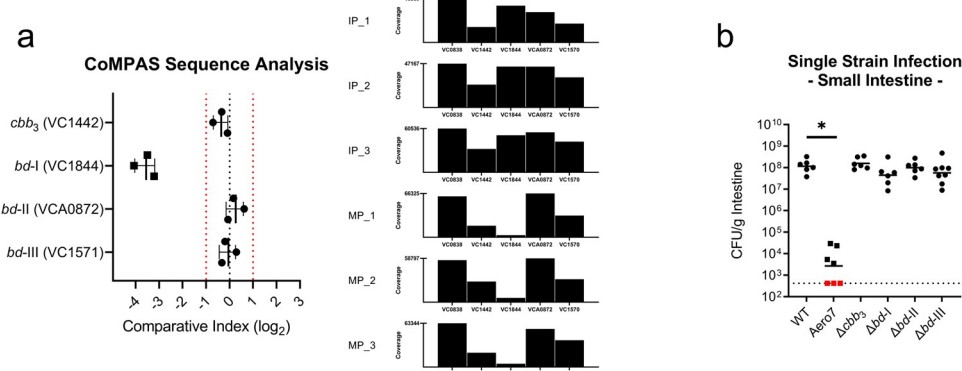

**Fig 6. *bd*-I oxidase is critical in competitive infection of the infant mouse small intestine, yet colonization is supported by functional redundancy of terminal oxidases in single strain infections.** Single oxidase *in vivo* colonization dynamics in the small intestine. (a) Comparative Multiplex PCR Amplification Sequencing (CoMPAS) sequence analysis. Comparative index scores were calculated as [(Output Pool$_{\text{Target Reads}}$ / Output Pool$_{toxT \text{ Reads}}$) / (Input Pool$_{\text{Target Reads}}$ / Input Pool$_{toxT \text{ Reads}}$)]. Vertical red dashed lines indicate a 2-fold change in output to input sequence ratios. Sequence coverage for each input pool (IP) and associated mouse output pool (MP) are shown in the bar plots. (b) Individual oxidase deletion single strain infections. Bars represent the geometric mean. Horizontal dashed lines indicate the limit of detection (LOD) and red dots indicate recovered CFUs were below the LOD. Statistical analysis was performed using GraphPad PRISM. *, $P < 0.05$. A Mann-Whitney U-test was used in the determination of significance between WT and Aero7 whereas an Analysis of Variance with *post hoc* Dunnett's multiple comparisons test was conducted on log transformed CFU/g intestine for all other strain comparisons.

(CoMPAS)). Individual terminal oxidase MuGENT strains were pooled along with a wild type strain in equal proportions, and served as the inoculum to infect infant mice. The small intestines were pooled from 6 mice and genomic DNA extracted along with the input inoculum for CoMPAS analysis.

Mutant allele abundances were determined for each oxidase complex and relative sequence abundances were normalized to *toxT* gene amplification (Fig 6A). Sequence coverage for each pool are presented in S2 Table. Comparative index scores are reported for each of the primary oxidase subunits VC1442 (*cbb₃*), VC1844 (*bd*-I), VCA0872 (*bd*-II), and VC1571 (*bd*-III). Comparative index scores were calculated as [(Output Pool$_{\text{Target Reads}}$ / Output Pool$_{toxT \text{ Reads}}$) / (Input Pool$_{\text{Target Reads}}$ / Input Pool$_{toxT \text{ Reads}}$)]. In this experiment, $^{\text{Mu}}bd$-I oxidase knockout strain was underrepresented in the output sequencing pool approximately 10-fold relative to the input. Conversely, all other MuGENT oxidase mutant strains were within a 2-fold change relative to the input. Overall, the *bd*-I oxidase was at a competitive disadvantage in the pooled infection and was determined to be the most important oxidase complex supporting growth *in vivo*.

**Single oxidase complex deletion infections.** Isogenic terminal oxidase deletion strains were also examined for colonization levels in single strain infections of the infant mouse. Loss of any one of the terminal oxidases did not result in a colonization defect in the small or large intestine of the infant mouse (Figs 6B and S6E). Wild type and mutant strains colonized to approximately $10^8$ CFU/g intestine. Counter to the observed defect in a $^{\text{Mu}}bd$-I knockout strain in our CoMPAS analysis, no colonization defect was present in single strain infections. These findings indicate that in single strain infections, loss of the *bd*-I oxidase was not detrimental to colonization and that likely functional redundancy exists among the oxidases that can support aerobic respiration during noncompetitive infection. We reasoned that a *bd*-I oxidase deficient strain was capable of colonizing the infant mouse, however, as cultures prepared anaerobically were delayed for aerobic growth as in Fig 2A, this growth delay resulted in the *bd*-I deficient strain being outcompeted during the pooled mouse infection. As multiple

oxidases were shown to support aerobic growth of *V. cholerae* in our growth assays, we hypothesized that the $cbb_3$, *bd*-I, and potentially the *bd*-III oxidase could all be supporting growth *in vivo*. This hypothesis also reflects data present in a large Tn-Seq dataset where transposon insertions into the $cbb_3$, *bd*-I, and *bd*-III oxidases showed a reduced capacity for colonization [42].

For all oxidase mutants, one concern is that disruption to oxidase function may impact other requirements for colonization such as virulence factor production or protection against reactive oxygen species (ROS). To determine whether mutations in the oxidases alter production of virulence factors required for colonization, TcpA protein levels in the mutants were examined as described in S1 Methods and were equivalent to wild type production (S7A–S7C Fig). As oxidative stress can also prevent bacterial growth *in vivo* and the *bd* oxidases of *Escherichia coli* exhibit low levels of catalase activity [43], we tested the minimum inhibitory concentraion of hydrogen peroxide on *V. cholerae* oxidase mutants, observing no growth defects for any mutant strain (S7D Fig). These findings support our conclusions that observed colonization defects can be attributed to a reduction in respiratory energy generation and not related to virulence factor production or increased ROS sensitivity that could have also limited colonization efficiency.

## Determining functionally redundant oxidases during infection

To identify which oxidases primarily support growth *in vivo*, triple oxidase isogenic deletion strains were used to colonize the infant mouse. By infecting with triple mutants, we could determine the importance of the single remaining oxidase. Strain +*bd*-I expressing solely the *bd*-I oxidase colonized comparable to wild type and strain +$cbb_3$ with a functional $cbb_3$ oxidase colonized at a ~1.5-fold reduction compared to wild type (Fig 7A). This finding further

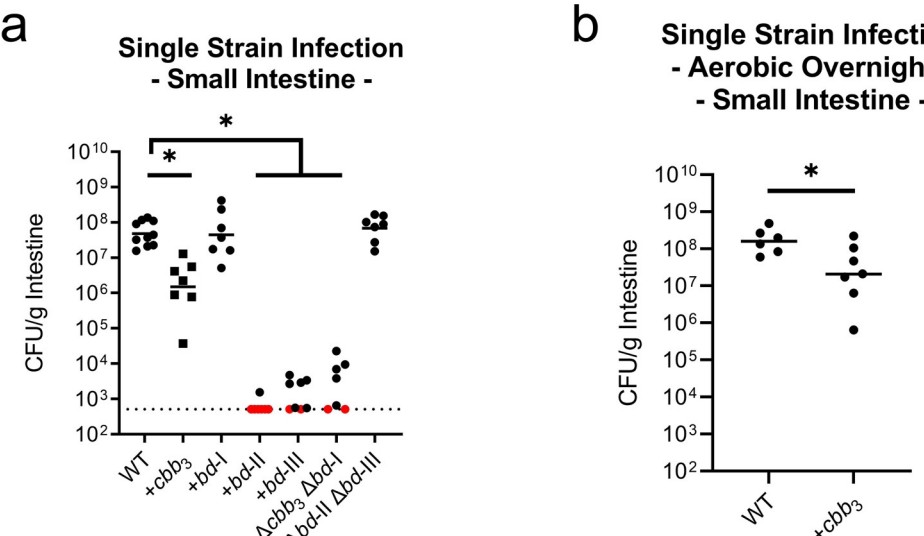

**Fig 7. *bd*-I oxidase alone supports wild type levels of colonization in the infant mouse small intestine with $cbb_3$ supporting colonization to a lesser extent.** (a) Combinatorial oxidase deletion i*n vivo* colonization dynamics in the small intestine. (b) Colonization of aerobically prepared wild type and +$cbb_3$ oxidase inoculums in the small intestine. Triple deletion mutant strains have a '+' with an oxidase name (e.g. +$cbb_3$), indicating the sole remaining oxidase, with the other three oxidases disrupted by mutation. Bars represent the geometric mean. Horizontal dashed lines indicate the limit of detection (LOD) and red dots indicate recovered CFUs were below the LOD. Statistical analysis was performed using GraphPad PRISM. *, $P < 0.05$. A Mann-Whitney U-test was used in the determination of significance between WT and +*bd*-II, +*bd*-III, and $\Delta cbb_3\Delta bd$-I whereas an Analysis of Variance with *post hoc* Dunnett's multiple comparisons test was conducted on log transformed CFU/g intestine for all other strain comparisons. *, $P < 0.05$.

supports *bd*-I oxidase as the primary oxidase of *V*. *cholerae*. Strains containing solely the *bd*-II or *bd*-III oxidase were unable to colonize (Fig 7A). This pattern of colonization was also reflected in the large intestine, however with higher levels of recovered CFU/g intestine, again likely due to the more anaerobic environment (S6F Fig). As the $cbb_3$ oxidase was determined to be less expressed and inactive in anaerobic conditions (Figs 2G and S2B), and mouse inocula were prepared anaerobically prior to infection, we investigated whether preparation in aerobic conditions could prime expression and activation of the $cbb_3$ oxidase to better support growth *in vivo*. However, despite preparing $+cbb_3$ oxidase cultures aerobically, we still observed a significant ~1-log reduction in colonization (Fig 7B). The colonization efficiency of aerobically grown $+cbb_3$ cultures was improved compared to anaerobically prepared $+cbb_3$ cultures but was not enough to support wild type levels of colonization. No significant difference was detected in the large intestine, which was also previously observed in the anaerobically prepared culture infections (S6G Fig).

As the $+bd$-III oxidase strain grew *in vitro* in aerobic LB conditions following aerobic overnight preparation while lacking both $cbb_3$ and *bd*-I oxidases, we looked to determine whether wild type expression levels of *bd*-III oxidase could support aerobic growth in this strain. By relative quantification of gene expression using an elongated Pfaffl method, $+bd$-III aerobic growth was found to be supported by increased expression of *bd*-III transcript, minimally 80x higher than wild type (S8A Fig). We hypothesized that this increased expression may be due to selective pressure for mutations that increase *bd*-III expression. A variant strain, $+bd$-III$^V$ was isolated that supported high levels of aerobic growth, had no defect for anaerobic growth, and exhibited increased expression of the *bd*-III oxidase (S8B–S8D Fig). We determined the genome sequence of this variant, identifying a mutation in *chrR* (VC2301), encoding an anti-sigma factor for SigmaE, which controls *bd*-III expression (S1 Table) [44]. In infant mouse infections, $+bd$-III$^V$ colonized near one order of magnitude lower than wild type in the small intestine of the infant mouse and was comparable to wild type in the large intestine (S8E and S8F Fig). Comparatively in infant mouse infections, the $+bd$-III$^V$ strain performed significantly better than $+bd$-III in the initial colonization infections. In wild type *V*. *cholerae*, we found that the *bd*-III oxidase is not typically expressed and does not contribute to *in vivo* colonization, however, in scenarios where the *bd*-III oxidase is the sole remaining oxidase and is expressed, it can support aerobic growth and *in vivo* colonization.

## Discussion

Terminal oxidases $cbb_3$, and particularly *bd*-I, serve as the major terminal reducing complexes to support the oxidative respiration required for population expansion of *V*. *cholerae* in the infant mouse. This is the first instance where a *bd*-type oxidase was determined to be the primary oxidase supporting growth of a Gram-negative bacterial pathogen in both atmospheric and *in vivo* oxygen environments. In *V*. *cholerae*, the *bd*-I oxidase facilitated aerobic growth and was critical for a rapid shift to aerobic respiration metabolism when transitioning from an anaerobic to aerobic environment. The finding that the $cbb_3$ and *bd*-I oxidases of *V*. *cholerae* are necessary for aerobic respiration in the low oxygen environment of the small intestine aligns with the typically low $K_m$ observed for each of these oxidase classes [19–21,45–47]. Our findings highlight that the low oxygen level in the small intestine is sufficient and essential for *V*. *cholerae* growth and implicates oxygen as a key electron acceptor for bacterial pathogenesis in the gut.

Investigating oxygen availability *in vivo* over the course of infection is important for understanding oxygen dynamics that shape *V*. *cholerae* pathogenesis. In *V*. *cholerae* infant rabbit infection, signatures of aerobic metabolism were observed as TCA cycle gene expression was

upregulated due to the presence of cholera toxin [48], indicating the cholera toxin may induce oxygen influx into the intestinal lumen. Gut microbial succession following *V. cholerae* infection also suggests conversion to a more oxygenated gut during the course of disease as facultative anaerobes predominate following human infection and display increased transcriptional abundance of the high-affinity $cbb_3$ oxidase during infection that became less abundant in the recovery period [49]. However, the direct relationship between *V. cholerae* and its cholera toxin on luminal oxygen availability has yet to be explored. In newly emergent strains of *V. cholerae*, intestinal cell damage has been found to occur with elevated bacterial loads in both mouse and rabbit models compared to previous pandemic strains [50]. Emergent strains capable of eliciting and withstanding inflammatory conditions of the intestine may benefit from increased oxygen availability, supporting enhanced population expansion. The findings of these studies support the conclusions made from this work, however, direct measurement of oxygen concentrations over the time course of disease remains to be determined. We find high conservation (99%-100%) among *V. cholerae* strains for all oxidases (S5 Table). Looking at closely related Vibrionaceae, we find divergent genomic sequences for the $cbb_3$ and *bd*-I oxidases and complete absence of the *bd*-II and *bd*-III oxidases complexes. This suggests that the $cbb_3$ and *bd*-I oxidases are widely used to support aerobic growth among Vibrionaceae, but *bd*-II and *bd*-III are unique to the *V. cholerae* lineage, and the evolutionary rationale for their presence is not yet understood.

In addition to the essential role of oxygen in growth and proliferation of *V. cholerae*, oxygen also serves as a signaling molecule regulating virulence gene expression in differentially oxygenated environments. *V. cholerae* maintains thiol-based switch regulators AphB and OhrR along with a two-component sensor ArcAB that respond to the presence of oxygen. Cysteine disulfide linkages that form within AphB and OhrR in reducing environments, such as the more anaerobic gut lumen, may prime *V. cholerae* cells for attachment by upregulating expression of the toxin coregulated pilus [51]. ArcAB also responds to oxygen, although indirectly through the action of redox active quinone electron carriers [52]. ArcA is typically activated in anaerobic conditions, yet in an experimental setup that favored microaerobic conditions, ArcA was needed for optimal virulence gene expression in the Classical biotype [53]. The contribution of ArcA to El Tor *V. cholerae* colonization and pathogenicity has yet to be explored. There may be different mechanisms of ArcA regulation between Classical and El Tor *V. cholerae*, as the Classical biotype is incapable of producing cholera toxin under anaerobiosis [54] whereas the El Tor biotype can produce cholera toxin anaerobically [6]. Oxygen sensing may also drive a chemotactic response in *V. cholerae* through aerotactic chemoreceptors Aer1 [55] and Aer2 [56], however this oxygen response pathway is unlikely to be required *in vivo* as non-chemotactic *V. cholerae* nevertheless colonize the infant mouse intestine [57].

Oxygen is also needed for growth and pathogenicity of other gastrointestinal pathogens including *E. coli* [58], *S.* Typhimurium [36], *L. monocytogenes* [34], and *C. jejuni* [59]. Infection with these pathogens leads to epithelial damage and gastroenteritis, which correlates with increases in luminal oxygen availability [60]. This influx of oxygen in the case of *E. coli* and *S.* Typhimurium further drives proliferation of these pathogens, exacerbating disease. In contrast, cholera is not typically characterized as an inflammatory disease, lacking gross pathological damage on host tissue, although *V. cholerae* does induce inflammatory markers in murine bone marrow-derived macrophages [61]. A question that emerges from our work is whether *V. cholerae* has a mechanism for driving oxygen into the gut to enhance growth and proliferation as do other intestinal bacterial pathogens. Our observation that oxidative respiration is critical for pathogen growth in the absence of tissue invasion, inflammation, and intestinal destruction is a novel finding in bacterial pathogenicity, although whether other microbes that

thrive on inflammation-induced electron acceptors can acquire oxygen during infection prior to when inflammation occurs is not clear.

Our work demonstrates oxidative respiration as essential to the replicative fitness of *V. cholerae* during infection. Despite the overall low concentrations of oxygen within the intestinal space, replication of *V. cholerae* is supported almost entirely by oxidative respiration to colonize the infant mouse small intestine. As oxygen in the human intestine is modulated by interactions between the host tissue and commensal microbial populations [36], new disease models of *V. cholerae* infection would be beneficial to investigate gut oxygen perturbations during disease. To enable colonization, *V. cholerae* mouse models require either the naïve intestine of the neonate with its limited microbiota, or antibiotic treatment of the adult [62,63]. In both instances, oxygen levels are predicted to be elevated compared to steady-state physiological hypoxia in the mature adult intestine [64,65]. Oxygen availability in these models, and likely in human infection, is therefore positively correlated to successful colonization of the intestine. Monitoring oxygen concentrations *in vivo* over the course of infection is challenging but may be important for understanding oxygen dynamics that shape *V. cholerae* pathogenesis.

## Materials and methods

### Ethics statement

All animal experiments in this study were conducted in accordance with all necessary regulations and requirements and was approved by the Institutional Animal Care and Use Committee at Michigan State University (PROTO201900421). Per the IACUC approved protocols, dam mice were euthanized by carbon dioxide inhalation and secondary cervical dislocation, and neonatal mice were euthanized by lethal dose of isoflurane and secondary cervical dislocation.

### Bacterial strains and growth conditions

Bacterial strains used in this study are listed in S3 Table. *Vibrio cholerae* El Tor C6706 was used as the wild type strain in this study and served as the strain background for all *V. cholerae* mutant derivatives. Strains were grown primarily on LB agar and used to inoculate 4ml LB media in preparation for subsequent assays. Antibiotics when required were in given concentrations: streptomycin (100μg/ml), spectinomycin (200μg/ml), and ampicillin (100μg/ml). Strains were grown at 37°C for all growth assays. Aerobic growth assays were performed at atmospheric oxygen concentrations whereas anaerobiosis for anaerobic growth was maintained using a Coy anaerobic chamber. For microaerobic growth conditions, an atmosphere of 5% oxygen, 10% carbon dioxide, 85% nitrogen was used. Deoxygenated LB media used in indicated growth assays was prepared by setting loose capped media in the Coy anaerobic chamber for a duration of 16-24h.

### MuGENT mutant strain construction

MuGENT generated mutant strains were constructed using Enhanced Multiplex Genome Editing by Natural Transformation [25]. Linear segments of *V. cholerae* genomic DNA were amplified using a primer with intentional base changes designed to introduce a frameshift mutation, removal of ATG start codon, insertion of 3-frame stop codons, and offsetting of the ribosomal binding site while also inserting a universal primer binding site. These fragments, along with a fragment containing an antibiotic resistance cassette in pseudogene VC1807 were transformed into a *V. cholerae* ΔrecJΔxseA pMMB-tfoX strain. Once all mutants were

integrated into the genome of carrier strains, a more traditional MuGENT approach [27] was used to amplify ~2Kb arms of homology on either side of the mutated site in each carrier strain to introduce into wild type *V. cholerae* by natural transformation, which maintains functional *recJ* and *xseA*. Candidate colonies were screened via colony PCR for target loci in a multiplex PCR reaction and confirmed by screening purified genomic DNA of each isolate. Strains were serially passaged in LB media to cure the pMMB-tfoX plasmid, where cured strains became sensitive to ampicillin 100μg/ml. Primers used to generate and confirm MuGENT mutant strains are listed in S4 Table.

### Isogenic deletion mutant strain construction

To confirm the phenotypes associated with MuGENT generated strains, isogenic deletion strains were also constructed. Growth phenotypes of isogenic deletion strains largely phenocopied MuGENT generated strains, thus we anticipated no off-target effects as equivalent growth was observed between mutant strains generated by two distinct DNA editing techniques. Whole genome sequencing was also used to validate select mutant strain genomes (See S1 Methods). Isogenic deletion strain constructs were generated using the positive allelic exchange vector pKAS32 [26]. Plasmid constructs were generated by first amplifying and purifying 1Kb DNA fragments upstream and downstream of target loci that contain homology base pairing to pKAS32. The pKAS32 vector was isolated from *E. coli* pKAS32 cultures using a QIAprep Spin Miniprep Kit (Qiagen) and digested with SacI and XbaI. DNA fragments and digested pKAS32 backbone were combined using Gibson Assembly (New England Biolabs) and transformed into *E. coli* ET12567 *ΔdapA* diaminopimelic acid auxotroph mating strain. Newly formed pKAS32 constructs were sequenced and correct vectors conjugated into *V. cholerae*. *V. cholerae*-pKAS32 strains were outgrown in LB at 37˚C 210rpm and subjected to >2500μg/ml streptomycin to select for strains that have excised the plasmid from its genome. Candidate mutant strains were screened by colony PCR and confirmed by screening purified genomic DNA. Primers to generate and confirm pKAS32 deletion strains are listed in S4 Table.

### *V. cholerae* terminal oxidase strain growth curves

Bacterial strains were grown either aerobically or anaerobically on LB streptomycin (100μg/ml) agar media and after 16-18h used to inoculate 4ml LB media. In aerobic conditions, 4 ml cultures were grown shaking at 210rpm, 37˚C whereas in anaerobic conditions 4 ml cultures were grown at 37˚C static. After 16h, bacterial strains were concentrated to a 1.0 $OD_{600}$. 700μl LB media was inoculated 1:1000 (0.7μl) with the 1.0 $OD_{600}$ resuspensions, vortexed, and aliquoted in triplicate 200μl volumes in a 96-well plate. Optical density was recorded every hour for the duration of the growth curve. Deoxygenated LB was used for anaerobic growth and benchtop LB used for aerobic growth.

### *V. cholerae* terminal reductase strain growth curves

Bacterial strains were grown on LB streptomycin (100μg/ml) agar media and after 16-18h used to inoculate 4ml LB media. After 16h, bacterial strains were concentrated to a 1.0 $OD_{600}$. 700μl LB media was inoculated 1:1000 (0.7μl) with the 1.0 $OD_{600}$ resuspensions, vortexed, and aliquoted in triplicate 200μl volumes in a 96-well plate. Optical density was recorded every hour for the duration of the growth curve. Deoxygenated LB was used for anaerobic growth curves and benchtop LB used for aerobic growth curves. Concentrations of alternative electron acceptors supplemented to LB media were as follows: 50mM sodium fumarate (Sigma), 50mM trimethylamine-N-oxide (TMAO) (Sigma), 50mM sodium nitrate (Sigma), and 50mM dimethyl

sulfoxide (DMSO) (Sigma). For strains grown in 50mM LB Nitrate media, after 3h, 5μM sodium hydroxide (Fisher Chemical) final concentration was added to alkalinize the growth media to support continued nitrate respiration.

## Wild type aerobic, microaerobic, and anaerobic RNA isolation and real-time quantitative PCR (RT-qPCR)

For each growth condition (aerobic / microaerobic / anaerobic) 4ml of LB media was placed in each environment at 37˚C to temper the media prior to inoculation in an effort to equalize the oxygen content and to pre-warm the media. Media was inoculated 1:1000 (4μl) with a 1.0 $OD_{600}$ wild type *V. cholerae* inoculum and grew shaking 210rpm for the aerobic culture and static for both microaerobic and anaerobic cultures. After 4h, culture tubes were centrifuged 4000rpm, 4˚C, for 10min and cell pellets were resuspended in 1ml TRIzol (Invitrogen). RNA was isolated from TRIzol suspensions using an RNeasy kit (Qiagen) coupled with an on-column DNase digestion (Qiagen) and Turbo DNase digestion (Invitrogen). RNA concentrations were measured with a UV/VIS Spectrophotometer and visualized on a 2% agarose gel.

cDNA was generated from RNA using Superscript III reverse transcriptase (Thermo Scientific). RT-qPCRs were performed using SYBR green master mix (Applied Biosystems) with 5ng of cDNA. Primers used to detect *recA* (VC0543), $cbb_3$ (VC1442), *bd*-I (VC1844), *bd*-II (VCA0872), and *bd*-III (VC1571) are listed in S4 Table. Relative quantification of oxidase expression was internally normalized to *recA* as the gene of reference and reported as relative fold change to *bd-III* oxidase expression in anaerobic conditions which served as the comparator target for analysis. Fold change was calculated using an elongated Pfaffl method for gene expression analysis, which is described in detail in S1 Methods [66].

## Infant mouse colonization assays

Infant mice were infected as described previously [67]. Briefly, three- to five-day-old mouse neonates (Charles River, Wilmington, MA) were orogastrically infected with approximately $10^6$ bacterial cells following 2 hours of separation from dam mice and maintained at 30˚C for 20h. After 20h, mice were euthanized, and intestinal segments weighed and homogenized in 4ml phosphate buffered saline (PBS). Intestinal homogenates were serially diluted and plated for CFU counts.

For single strain infections, dilutions were plated on LB streptomycin (100 μg/ml) for growth and enumeration. For MuGENT strain competition assays, dilutions were additionally plated on LB spectinomycin (200 μg/ml) for differentiation from co-infected wild type *V. cholerae*. Plates were consistently incubated at 37˚C in a Coy anaerobic chamber to ensure growth of oxidase deletion strains.

## CoMPAS infant mouse infection and sequencing

Wild type, $^{Mu}cbb_3$, $^{Mu}bd$-I, $^{Mu}bd$-II, and $^{Mu}bd$-III MuGENT strains were combined in equal ratios and a total of 6 infant mice were infected with a final 0.01 $OD_{600}$ inoculum, approximately ~$10^6$ CFU. Remaining inoculum volume was spun down at 4˚C, 4000rpm, for 10min and DNA of the inoculum pool (IP) was isolated using a QIAamp PowerFecal Pro DNA Kit (Qiagen). After 20h infection, small intestinal segments were homogenized and pooled from the 6 infant mice. Pooled homogenates were filtered on ice using a 70μm filter to remove residual intestinal tissue. From here, DNA of the mouse pool (MP) was isolated using a QIAamp PowerFecal Pro DNA Kit (Qiagen).

Recovered DNA samples were normalized to 100ng/μl using Qubit dsDNA HS analysis and subsequently, multiplex amplicon sequencing was performed on both inoculum and mouse

pools. Amplification targets included the primary subunit of each MuGENT oxidase complex (VC1442, VC1844, VCA0872, and VC1571) as well as *toxT* (VC0838) which is present in all input strains. Primers used are listed in S4 Table, amplification was carried out for 30 cycles. PCR products were purified using the QIAquick PCR Purification Kit (Qiagen) and quantified by Qubit dsDNA HS and normalized to 20ng/µl for MiSeq amplicon sequencing by the MSU RTSF Genomics Core (Michigan State University). Sequencing was conducted on a MiSeq Nano v2 flow cell using a 2x250bp paired end format. Sequence barcodes were trimmed and target loci read counts were quanitified using Geneious software.

### *In vitro* competition assays

Bacterial strains were grown in 4ml LB media for 16-18h and resuspended to 1.0 OD$_{600}$. Wild type and an individual target strain were combined in a 1:1 ratio and a 1:1000 (4µl) dilution used to inoculate deoxygenated LB for anaerobic competitions or benchtop LB for aerobic competitions. Anaerobic competitions were grown at 37°C static and aerobic competitions were grown 37°C 210rpm shaking. After 20h of growth, cultures were serially diluted and plated for colony forming units. WT vs. Aero7 and WT vs. Ana4 competitions were plated on LB streptomycin (100µg/ml) and LB spectinomycin (200µg/ml) to determine strain ratios. Individual deletion strain competitions were plated on LB streptomycin (100µg/ml) plus 5-bromo-4-chloro-3-indolyl β-D-galactopyranoside (X-Gal; 40µg/ml) for blue-white screening to determine strain ratios between a target strain and a Δ*lacZ* strain that served as a wild type comparison in the competition assays. Individually grown cultures followed the same growth procedure, however strains were not combined with Δ*lacZ* strain and were recovered on LB streptomycin (100µg/ml) for enumeration.

## Supporting information

**S1 Methods. Supplemental Materials and Methods.**
(PDF)

**S1 Fig. MuGENT spectinomycin selective marker shows no fitness defect *in vitro*.** Comparison of wild type and Δ*lacZ* C6706 *V. cholerae* strains with and without MuGENT spectinomycin selective marker in pseudogene VC1807. (a) Aerobic i*n vitro* competition assay after 20h. (b) Aerobic growth curve assay. (c) Anaerobic *in vitro* competition assay after 20h. (d) Anaerobic growth curve assay. All assays were performed in LB media. Bars in *in vitro* competitions represent the arithmetic mean where error bars represent the standard error of the mean. Growth curves are an average of three biological replicates where error bars represent the standard error of the mean.
(TIF)

**S2 Fig. *cbb*$_3$ deficient strains and wild type grown anaerobically do not maintain a functional *cbb*$_3$ oxidase complex.** *V. cholerae* cultures were grown on LB agar plates and spotted onto a rapid test DrySlide containing N$_1$N$_1$N'$_1$N'-tetramethyl-*p*-phenylene-diamine dihydrochloride (Wurster's blue; TMPD) that turns blue when reduced by cytochrome *c* oxidases. This colorimetric detection assay indicates active cytochrome *c* oxidases but does not provide quantitative measurements of complex activity. (a) *V. cholerae cbb*$_3$ mutant strain spots and *E. coli* (cytochrome *c* deficient) control. (b) Wild type *V. cholerae* grown in aerobic, microaerobic, and anaerobic conditions.
(TIF)

**S3 Fig. *V. cholerae* oxidases generally support the same pattern of growth in minimal M9 0.2% D-glucose media as seen in LB and combinatorial MuGENT mutants closely**

**recapitulate triple isogenic deletion mutant growth in LB.** (a-b) Single terminal oxidase deletion mutants grown in M9 0.2% D-glucose, aerobically and anaerobically, respectively. Inoculums for all growth experiments were prepared in anaerobic conditions. (c-d) Combinatorial terminal oxidase deletion mutant growth in LB. Inoculums were prepared anaerobically and subsequently grown in aerobic and anaerobic conditions, respectively. Triple deletion mutant strains have a '+' with an oxidase name (e.g. $+cbb_3$), indicating the sole remaining oxidase, with the other three oxidases disrupted by mutation. Growth curves are an average of three biological replicates where error bars represent the standard error of the mean. (TIF)

**S4 Fig. Individual deletion oxidase strains reach comparable colony forming unit counts at 20h.** *V. cholerae* individual deletion strain endpoint CFU in LB media at 20h. CFU endpoints were determined in (a) aerobic and (b) anaerobic growth conditions. CFU's reported are an average of three biological replicates where error bars represent the standard error of the mean. (TIF)

**S5 Fig. Terminal reductase mutants are variable for aerobic growth in the presence of cognate electron acceptor molecules.** *V. cholerae* terminal reductase aerobic growth characteristics in LB in the presence and absence of alternative electron acceptors (a) 50mM fumarate, (b) 50mM trimethylamine-N-oxide (TMAO), (c) 50mM nitrate, and (d) 50mM dimethyl sulfoxide (DMSO). Inoculums were prepared aerobically. Closed symbols indicate LB growth media lacked an alternative electron acceptor whereas open symbols indicate LB growth media was supplemented with a given alternative electron acceptor. Growth curves are an average of three biological replicates where error bars represent the standard error of the mean. (TIF)

**S6 Fig. Functional terminal oxidases, but not alternative terminal reductases, are required for optimal colonization of the large intestine.** Aero7 and Ana4 colonization of the large intestine in both single strain and competition infections. (a) Single strain infection of strain Aero7. (b) Competition infection of strain Aero7. (c) Single strain infection of strain Ana4. (d) Competition infection of strain Ana4. Bars represent the geometric mean. Horizontal dashed lines indicate the limit of detection (LOD) and red dots indicate recovered CFUs were below the LOD. Competitive index scores were calculated as [(Mutant$_{Output}$/WT$_{Output}$) / (Mutant$_{Input}$/WT$_{Input}$)]. Statistical analysis was performed using GraphPad PRISM. *, $P < 0.05$. A Mann-Whitney U-test was used in the determination of significance between WT and Aero7. A Student's *t* test was performed on log transformed data in the determination of significance between WT and Ana4. Single strain *in vivo* colonization assays in the large intestine for (e) individual and (f) combinatorial oxidase deletion strains. (g) Single strain colonization of aerobically prepared wild type and $+cbb_3$ oxidase inoculums in the large intestine. Triple deletion mutant strains have a '+' with an oxidase name (e.g. $+cbb_3$), indicating the sole remaining oxidase, with the other three oxidases disrupted by mutation. Bars represent the geometric mean. Horizontal dashed lines indicate LOD, and red dots indicate recovered CFUs were below the LOD. Statistical analysis was performed using GraphPad PRISM. *, $P < 0.05$. A Mann-Whitney U-test was used in the determination of significance between WT and Aero7 and WT and $+bd$-III whereas an Analysis of Variance with *post hoc* Dunnett's multiple comparisons test was conducted on log transformed CFU/g intestine for all other strain comparisons. (TIF)

**S7 Fig. TcpA production is functional and hydrogen peroxide tolerance is equivalent among individual oxidase deletion mutants.** (a) Western blot visualization of TcpA, a

required virulence factor in *V. cholerae* pathogenesis, in both standard and anaerobic AKI conditions. (b) Densitometry analysis of TcpA production in standard AKI conditions. (c) Densitometry analysis of TcpP production in anaerobic AKI conditions. TcpA levels are displayed as relative to TcpA production in wild type cells. ImageJ was used to perform the densitometry analysis across three biological replicates. Horizontal bars represent the arithmetic mean where error bars represent the standard deviation of the mean. (d) Hydrogen peroxide minimum inhibitory concentration determination of individual oxidase deletion strains. Growth percentage was calculated as a function of optical density for test strains in various concentrations of $H_2O_2$ (0.15625mM, 0.3125mM, 0.625mM, 1.25mM, 2.5mM, 5mM, and 10mM) divided by the optical density for wild type *V. cholerae* grown in LB media without $H_2O_2$. All strains showed signs of growth reduction at 1.25mM $H_2O_2$ and were all entirely inhibited for growth at 2.5mM $H_2O_2$. Data points represent the arithmetic mean of three biological replicates with error bars representing the standard error of the mean.
(TIF)

**S8 Fig. Variant +*bd*-III strain (+*bd*-III$^V$) indicates that the *bd*-III oxidase, when expressed, is capable of supporting aerobic respiration in *V. cholerae* and colonization of the infant mouse.** (a) Wild type and +*bd*-III strain ($\Delta cbb_3$ $\Delta bd$-I $\Delta bd$-II) *bd*-III expression. Expression was determined for the primary subunit of the *bd*-III oxidase VC1571. Dots represent biological replicates of relative *bd*-III expression between +*bd*-III and wild type strains. Bars represent arithmetic mean with error bars representing the standard error of the mean. (b-c) Wild type and +*bd*-III$^V$ growth in LB. Inoculums were prepared aerobically and subsequently grown in aerobic and anaerobic conditions, respectively. Data points represent the mean of triplicate growth curves with error bars representing the standard error of the mean. (d) *In vitro* expression of terminal oxidases in +*bd*-III$^V$ strain. Expression was determined for the primary subunit of each oxidase complex (VC1442, VC1844, VCA0872, VC1571). Bars represent the arithmetic mean with error bars representing the standard error of the mean. (e) Single strain infection of +*bd*-III$^V$ in the small intestine. (f) Single strain infection of +*bd*-III$^V$ in the large intestine. Bars in single strain infections represent the geometric mean. Statistical analysis was performed using GraphPad PRISM. *, $P < 0.05$. A Student's *t* test was performed on log transformed data in the determination of significance between WT and +*bd*-III$^V$ in both the small and large intestine.
(TIF)

**S1 Table. Whole genome sequencing single nucleotide polymorphism analysis annotation.** (XLSX)

**S2 Table. CoMPAS sequencing reads.** (XLSX)

**S3 Table. Bacteria strain list.** (XLSX)

**S4 Table. Primer list.** (XLSX)

**S5 Table. Oxidase sequence comparison.** (XLSX)

## Acknowledgments

We thank Ankur Dalia, Ph.D. for helpful advice and strains that supported successful multiplex genomic editing of *V. cholerae* in this study.

## Author Contributions

**Conceptualization:** Andrew J. Van Alst, Victor J. DiRita.

**Data curation:** Andrew J. Van Alst.

**Formal analysis:** Andrew J. Van Alst, Lucas M. Demey.

**Funding acquisition:** Victor J. DiRita.

**Investigation:** Andrew J. Van Alst, Lucas M. Demey.

**Methodology:** Andrew J. Van Alst, Victor J. DiRita.

**Software:** Andrew J. Van Alst.

**Supervision:** Victor J. DiRita.

**Validation:** Andrew J. Van Alst, Lucas M. Demey.

**Writing – original draft:** Andrew J. Van Alst, Victor J. DiRita.

**Writing – review & editing:** Andrew J. Van Alst, Lucas M. Demey, Victor J. DiRita.

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
