## [Decision Letter · Decision Letter 0]

19 Dec 2021

Dear Dr. DiRita,

Thank you very much for submitting your manuscript "Oxidative respiration through the bd-I and cbb3 oxidases is required for Vibrio cholerae pathogenicity and proliferation in vivo." for consideration at PLOS Pathogens. As with all papers reviewed by the journal, your manuscript was reviewed by members of the editorial board and by three independent reviewers. 

Two of the reviewers advised minor revision, although some suggestions/concerns will require additional experiments. The third reviewer advised major revision, with some suggestions that also imply additional experiment(s). From our point of view, any of the points raised seem to be easily answered and/or executable, but, considering that they may require some additional experiment(s), these issues must be addressed before we would be willing to consider a revised version of your study. 

We cannot make any decision about publication until we have seen the revised manuscript and your response to the reviewers' comments. Your revised manuscript is also likely to be sent to reviewers for further evaluation.

Sincerely,

Nuno Miguel Simões dos Santos

Associate Editor

PLOS Pathogens

Nina Salama

Section Editor

PLOS Pathogens

Kasturi Haldar

Editor-in-Chief

PLOS Pathogens

orcid.org/0000-0001-5065-158X

Michael Malim

Editor-in-Chief

PLOS Pathogens

orcid.org/0000-0002-7699-2064

Reviewer's Responses to Questions

**Part I - Summary**

Reviewer #1: This paper reports the role of terminal oxidase in the survival and colonization abilities of V. cholerae. The authors analyzed the roles of cytochromes cbb3 and the bd and concluded that bd-I is the main oxidase in aerobic growth. They also found that these enzymes play important roles during colonization.

Some clarifications:

The authors should include data showing the growth rates of the different mutants.

The authors refer to the terminal oxidases of Pseudomonas aeruginosa. In P. aeruginosa, cytochrome aa3 which the most energy efficient enzyme is not the main oxidase expressed during aerobic and it is only expressed in very specific conditions. (starvation)

It is possible that V. cholerae uses bd-I as the “main oxidase” since the turnover of this enzyme is faster than that of cbb3.

The authors used a plate colorimetric assay to detect activity of cbb3. They need to make it clear that is a detection assay not a measurement.

Overall, this paper offer important insights of the role of energy metabolism during the life cycle of V. cholerae.

Reviewer #2: In this study, Andrew et al. investigate the function of multiple terminal oxidoreductases for aerobic and anaerobic respirations in Vibrio cholerae by using multiple knockout mutant strains. One of three bd-type quinol oxidases, bd-I, was identified to be the primary enzyme for aerobic growth and virulence to infant mouse. Anaerobic respiration was not important for infection in infant mouse.

The authors performed laborious work for construction of multiple mutants, competitive experiments, and infection assays. The obtained data seems to be plausible and important.

Reviewer #3: The authors discovered that two of the four oxygen-dependent respiration mechanisms are essential for V. cholerae to grow during infection. Especially, they constructed the V. cholerae strains lacking aerobic or anaerobic respiration using a technique called MuGENT. In addition, using CoMPAS, they showed that bd-1 and cbb3 oxidases are essential for colonization in the infant mouse.

**Part II – Major Issues: Key Experiments Required for Acceptance**

Reviewer #1: NO Major issues.

Reviewer #2: I have a concern about the aerobic culture conditions. The authors used 96-well plates for the aerobic growth assays, but oxygen supply might not be enough in the culture.

I recommend confirming the results under highly aerating conditions using shaking flask or jar fermenter.

Reviewer #3: Here are some comments and suggestions to improve the manuscript;

1. In this manuscript, as their title, the authors asserted that the oxygen dependent respiration is essential for V. cholerae pathogenicity and proliferation in vivo. Thus, they showed the colonization abilities of mutants in infant mouse intestinal tract. But they did not show any pathogenicity of mutants. For example, were any mice dead within 20h after the infection? Please present the survival rate of them. Or present data to confirm the pathogenicity of them.

2. The authors determined that the bd-1 and cbb3 are essential for colonizing the small intestine of infant mouse. It would be informative to show how well these two gene clusters are conserved in V. cholerae genomes.

3. Page 9, line 205-207. Authors concluded from the growth assay and the expression profiles that the bd-1 oxidase is critical for aerobic respiration in V. cholerae as it transits from an anaerobic to aerobic environment. However, in Figure 2G, authors presented the expression levels of terminal oxidase genes in aerobic, microaerobic, and anaerobic conditions which were not conducted in transiting manner from one to another as they mentioned. To make the demonstration more plausible, please present the gradients in expression levels of terminal oxidase genes under transition condition from an anaerobic to aerobic environment.

4. In Figure 2 C and D. From in vitro competition assay, they demonstrated a competitive growth defect of both cbb3 and bd-1 deficient strains in aerobic conditions. In a competition assay, please display each CFU of the wild-type and the mutants as controls by culturing them individually.

5. Did you use the anaerobic chamber for all the anaerobic experiments? In line 597, 618 and 670, it says that deoxygenated LB was used for anaerobic conditions. Please clarify whether the anaerobic culture conditions of the assay were all referring to the experiment inside the anaerobic chamber.

6. In case of microaerobic culture, what does the term ‘microaerobic’ refer to? What are the standards and how did you set up the microaerobic condition? Please elaborate more about the culture conditions. If O2 concentration was set to certain degree in microaerobic culture, please state it.

7. In line 639. They checked the CFU of intestinal homogenates to test the colonizing abilities of aerobic or anaerobic mutants. For this assay, in which condition did you culture the plates of the two mutants? If you culture them in aerobic conditions, it is too favorable for one of the two mutants, which is making the experiment biased. It could be comprehensively explained with the CFU data from the suggestion no.4.

**Part III – Minor Issues: Editorial and Data Presentation Modifications**

Reviewer #1: The authors should include data showing the growth rates of the different mutants.

The authors refer to the terminal oxidases of Pseudomonas aeruginosa. In P. aeruginosa, cytochrome aa3 which the most energy efficient enzyme is not the main oxidase expressed during aerobic and it is only expressed in very specific conditions. (starvation)

It is possible that V. cholerae uses bd-I as the “main oxidase” since the turnover of this enzyme is faster than that of cbb3.

The authors used a plate colorimetric assay to detect activity of cbb3. They need to make it clear that is a detection assay not a measurement.

Reviewer #2: There are two types in the bd-type quinol oxidases, i.e., the canonical bd-type and the cyanide-insensitive oxidase (CIO)-type. The CIO-type enzymes of Acetobacter and Pseudomonas species are reported to have low affinity to oxygen. bd-III of V. cholerae seems to be the CIO-type and might have high Km value. That might be the reason why the +bd-III grew faster when aerobic preculture was used (Fig. 3C).

Lines 79-80; The authors referred to oxygen reductases as terminal oxidase and reductases for alternative electron acceptors as terminal reductases. It might be better to define the terms “terminal oxidase” and “terminal reductase” here.

Lines 132, 136, and 164; Italicize “c” and “bc” of cytochromes c and bc1.

Figures 2, 3, and 4; Resolution of the graphs is very low, and it is hard to distinguish the lines.

Lines 169-170; Indicate that the anaerobic growth is due to anaerobic fermentation.

Lines 594-595; Shaking or aerating conditions for the aerobic cultivation should be indicated.

Sheet 2 of S3 Table and sheet 1 of S4 Table may be not related to this study.

Reviewer #3: 8. They stated ‘as the adult gut has greater anaerobic luminal volume compared to the limited luminal space in the infant’. Please add the reference.

PLOS authors have the option to publish the peer review history of their article (what does this mean?). If published, this will include your full peer review and any attached files.

Reviewer #1: **Yes: **Blanca Barquera

Reviewer #2: **Yes: **Hiroyuki Arai

Reviewer #3: No
---

## [Editor Report · Decision Letter 1]

5 Apr 2022

Dear Dr. Victor J.,

We are pleased to inform you that your manuscript 'Vibrio cholerae requires oxidative respiration through the bd-I and cbb3 oxidases for intestinal proliferation.' has been provisionally accepted for publication in PLOS Pathogens.

Best regards,

Nuno Miguel Simões dos Santos

Associate Editor

PLOS Pathogens

Nina Salama

Section Editor

PLOS Pathogens

Kasturi Haldar

Editor-in-Chief

PLOS Pathogens

orcid.org/0000-0001-5065-158X

Michael Malim

Editor-in-Chief

PLOS Pathogens

orcid.org/0000-0002-7699-2064
---

## [Editor Report · Acceptance letter]

27 Apr 2022

Dear Dr. DiRita,

We are delighted to inform you that your manuscript, "Vibrio cholerae requires oxidative respiration through the bd-I and cbb3 oxidases for intestinal proliferation.," has been formally accepted for publication in PLOS Pathogens.

Best regards,

Kasturi Haldar

Editor-in-Chief

PLOS Pathogens

orcid.org/0000-0001-5065-158X

Michael Malim

Editor-in-Chief

PLOS Pathogens

orcid.org/0000-0002-7699-2064